# The Ethylene Response Factor ERF5 Regulates Anthocyanin Biosynthesis in ‘Zijin’ Mulberry Fruits by Interacting with *MYBA* and *F3H* Genes

**DOI:** 10.3390/ijms23147615

**Published:** 2022-07-09

**Authors:** Rongli Mo, Guangming Han, Zhixian Zhu, Jemaa Essemine, Zhaoxia Dong, Yong Li, Wen Deng, Mingnan Qu, Cheng Zhang, Cui Yu

**Affiliations:** 1Academy of Agricultural Sciences, Industrial Crops Institute of Hubei, Wuhan 430064, China; monamorus@hbaas.com (R.M.); mrhan888@163.com (G.H.); zhuzhixian@hbaas.com (Z.Z.); dongzhaoxia9@hbaas.com (Z.D.); liyong8057@hbaas.com (Y.L.); dengwen@hbaas.com (W.D.); 2National Key Laboratory of Plant Molecular Genetics, CAS Center for Excellence in Molecular Plant Sciences, Chinese Academy of Sciences, Shanghai 200032, China; jemaa@picb.ac.cn (J.E.); qmn@cemps.ac.cn (M.Q.)

**Keywords:** anthocyanin, antioxidant, ethylene-responsive factors (ERF), *F3H*, metabolism, mulberry, *MYBA*, transcriptomes

## Abstract

Ethylene promotes ripening in fruits as well as the biosynthesis of anthocyanins in plants. However, the question of which ethylene response factors (ERFs) interact with the genes along the anthocyanin biosynthesis pathway is yet to be answered. Herein, we conduct an integrated analysis of transcriptomes and metabolome on fruits of two mulberry genotypes (‘Zijin’, ZJ, and ‘Dashi’, DS, with high and low anthocyanin abundance, respectively) at different post-flowering stages. In total, 1035 upregulated genes were identified in ZJ and DS, including *MYBA* in the MBW complex and anthocyanin related genes such as *F3H*. A KEGG analysis suggested that flavonoid biosynthesis and plant hormone signaling transduction pathways were significantly enriched in the upregulated gene list. In particular, among 103 *ERF* genes, the expression of *ERF5* showed the most positive correlation with the anthocyanin change pattern across both genotypes and in the post-flowering stages, with a Pearson correlation coefficient (PCC) of 0.93. Electrophoresis mobility shift assay (EMSA) and luciferase assay suggested that ERF5 binds to the promoter regions of MYBA and F3H and transcriptionally activates their gene expression. We elucidated a potential mechanism by which ethylene enhances anthocyanin accumulation in mulberry fruits and highlighted the importance of the *ERF5* gene in controlling the anthocyanin content in mulberry species. This knowledge could be used for engineering purposes in future mulberry breeding programs.

## 1. Introduction

Mulberry (*Morus alba* L.) is a deciduous tree belonging to the Moraceae family. It is regarded as an eminent economic plant; its foliage can be used in sericulture as a basic feedstock for the monophagous silkworm [1]. The leaves and fruit extracts of *Mallotus pellatus* have been extensively reported to promote health, e.g., inhibiting human LDL oxidation and preventing atherosclerosis in apolipoprotein E-deficient mice [2,3]. Mulberry has other important uses, e.g., as a fruit with high nutritional value and appealing adaptability [4].

A number of anthocyanins extracted from mulberry fruits apparently give rise to the differential fruit colors in different genotypes [5]. The anthocyanin biosynthetic pathway begins with the chalcone synthase (CHS) mediating naringenin chalcone synthesis from 4-coumaroyl-CoA and malonyl-CoA. Then, naringenin chalcone can be isomerized by chalcone isomerase (CHI). Among the naringenin family, cyanidine-3-O-glucoside and cyanidine-3-O-rutinoside are the major anthocyanin components in mulberries [6]. Recent research on mulberry anthocyanin production has focused mostly on defining and/or profiling some genes which are thought to be involved in the biosynthesis of anthocyanins [1,7].

Notably, the anthocyanin biosynthetic pathway constitutes an extension of the global flavonoid pathway, which is mainly regulated by the MYB–bHLH–WD40 (MBW) transcription complexes [8] and the ethylene response factor (ERF) genes during the fruit ripening process [9]. The former complex (MBW) includes R2R3-myeloblastosis (MYB), basic helix-loop-helix (bHLH) and WD40 proteins. Notably, different groups of MBW transcription complexes were reported in the same species. For example, in Arabidopsis, four MYBs subgroup, three IIIf bHLHs subgroups and a TTG1 protein have been characterized [10]. It has also been reported that in mulberry fruits, bHLH3 represents a key modulator for the MBW metabolic network regulator, functioning as a form of negative feedback to balance the anthocyanin content and proanthocyanidins accumulation [11]. It was recently reported that ethylene promotes fruit ripening in several species, e.g., pear [12] and tomato [13], leading to an early accumulation of anthocyanin. However, few insights are available about the functional molecular mechanism of *ERF* genes in the anthocyanin biosynthetic pathway during mulberry fruit ripening.

Multi-omics analysis has emerged as a powerful tool, facilitating in depth investigations into the plant kingdom genome. This tool is dedicated to dissecting the metabolic-transcriptomic profiles of plants by encompassing various molecular biology approaches. For instance, metabolic analyses constitute an effective and quantitative method to elucidate the regulatory mechanisms of important metabolic pathways, as reported in maize [14] and alfalfa [11]. Integrating transcriptomic analysis and metabolism is strongly recommended in attempts to discover new genes and key biological pathways which would allow us to better dissect the anthocyanin biosynthesis mechanisms in mulberry fruits.

In this report, it was hypothesized that some *ERF* genes could bind to the promoter region of the genes involved in the anthocyanin biosynthetic process, and hence, could promote early anthocyanin accumulation in mulberry fruits. In this regard, we combined transcriptomes and metabolome analysis to investigate the biochemical compositions and molecular features of the fruits of two mulberry genotypes, i.e., ‘Zijin’ (ZJ) and ‘Dashi’ (DS), characterized by their varying anthocyanin contents. The expression of the *MYBA* gene in the MBW complex was found to be enhanced in the anthocyanin abundant genotype (ZJ). In addition, the abundances of *ERF* genes were qualitatively analyzed in both mulberry genotypes across different post-flowering stages and the key *ERF* genes were further validated in vitro using ethylene-treated fruit samples.

We propose a molecular mechanism of the key *ERF* genes in regulating the anthocyanin biosynthetic pathway. Understanding this mechanism could be helpful for engineering future mulberry genotypes for sericulture breeding programs. The novelty of this study lies in revealing how ERF5 transcriptionally activates *MYBA* and *F3H* under ethylene treatment, leading to an accumulation of anthocyanin in both mulberry genotypes, but to various extents.

## 2. Results

### 2.1. Morphological and Physiological Traits of Mulberry Fruits

The difference in the anthocyanin contents of the two mulberry genotypes was remarkable, especially at the three post-flowering stages (I, II and III), where ZJ was defined by its high anthocyanin level. In contrast, DS was observed to have a low anthocyanin content (Figure 1A). Thus, except for stage IV, our results show a significant difference in anthocyanin accumulation between the two mulberry genotypes during the three post-flowering stages (Figure 1A). Furthermore, we noticed a gradual increase in the fruit length and anthocyanin content with progress in the fruit developmental stage from 5 to 31 days after flowering (Figure 1B,C). We recorded a significant increase in the anthocyanin content for ZJ compared to that of DS at the latest stages, i.e., III and IV (Figure 1B). Additionally, the fruit length of the DS genotype was increased by up to 30% at the latest stages (III and IV) compared to that of ZJ (*p* < 0.05; Figure 1C). In addition, for both mulberry genotypes, the SSC and fruit solidity exhibited increasing and decreasing patterns, respectively, through the different post-flowering stages (5 to 31 days after flowering), but no significant difference between the two mulberry genotypes (DS and ZJ) was observed regarding these two parameters, i.e., SSC and fruit solidity (Figure 1D,E).

### 2.2. Quality Control of Mulberry Transcriptome Sequencing

In order to elucidate the changes in the molecular aspect of mulberry fruits between both genotypes (DS and ZJ), we performed a transcriptome analysis at four post-flowering stages. Notably, samples were sequenced using the Illumina Hiseq2000 technology system, generating 74,946,024 and 79,763,262 raw reads for DS and ZJ, respectively (Appendix A). The average length of a single read was ~150 bp (Appendix A). After filtering out the low-quality reads, 73,783,426 and 78,744,230 clean reads (which represent 99.51% and 99.63% of the raw reads sequences for DS and ZJ, respectively) were obtained with Q_20_ values of 97.79% and 97.80% for DS and ZJ, respectively (Appendix A). However, the error rates for the single nucleotide ranged from 0.02% to 0.06% for both genotypes (Appendix A).

The bases content (A, T, C and G) of the different nucleic acids was used to estimate the quality of the cDNA library construction, which showed a stable level in A/T and G/C contents across the different read fragments (Appendix A). Indeed, more than 97% clean reads were detected, with only 2% being of low quality and less than 0.004% non-detectable (Appendix A). Thus, the FPKM values maintained similar expression levels of transcripts across all of the studied samples (Appendix A). Nevertheless, the Pearson correlation coefficient (PCC) between samples showed high correlation (with R^2^ > 0.8) across the three investigated biological replicates for each mulberry genotype (Appendix A).

Afterwards, we analyzed the alternative splicing events based on the transcriptomes of both mulberry genotypes and found that the skipped exons encompassed around 7000 cases when comparing the samples of the two mulberry genotypes at different post-flowering stages (Appendix A). This represents a high level for both the alternative 3′ splice site and the alternative 5′ one, relative to rare cases of the retained introns in the mulberry genome.

### 2.3. Principal Component Analysis (PCA) and DEG Profiling in Both Mulberry Genotypes

The PCA performed on the transcriptomes data exhibited a clear clustering in the samples of the two mulberry genotypes (DS and ZJ) at the different post-flowering stages (Figure 2A). The score for the top 2 principal components was higher than 80% (Figure 2A). To gain more insights about the DEGs, and taking into account the developmental factors between both mulberry genotypes, we compared the DEGs for each genotype at the last three post-flowering stages (II, III and IV) relative to stage I (Figure 2B). Notably, 3945 and 3302 overlapped genes were identified over the three post-flowering stages for DS and ZJ, respectively (Figure 2B).

### 2.4. Determination of the Differential Molecular Profiling between the Two Mulberry Genotypes at Stage III

In this section, we mainly focus on investigating the abundance and expression of DEGs at post-flowering stage III, regarding the remarkably higher anthocyanin content in ZJ compared to DS at this stage (III). In this regard, our results revealed that 6166 genes were found to be overlapped between the two mulberry genotypes at stage III compared to stage I. Among them (6166 genes), we identified 1035 upregulated and 658 downregulated genes in ZJ compared to DS at this stage (Figure 2C). Thus, the abundance of the top 20% upregulated and downregulated genes was determined for both mulberry genotypes (Figure 2D).

Therefore, we performed GO enrichment and a KEGG analysis to gain more accurate and valuable insights about the potential biological pathways for the phenotypic differences between the fruits of the two mulberry genotypes (Figure 1A). In the list of upregulated genes at this stage, based on a GO analysis, we detected that some biological processes related to the secondary metabolism, lignin catabolic process, cell wall macromolecular biosynthesis, xylem biosynthesis, and oxido-reductase activity were significantly enriched (Figure 3A). Specifically, the reactions of these biological processes are known to occur across the cell membrane (Figure 3A). Concerning the list of downregulated genes at stage III, we determined that some biological pathways associated with transporter activity and transferase activity were significantly enriched (Figure 3B).

However, a KEGG analysis performed on the top-20 terms of the DEGs showed that some biological processes related to the secondary metabolites, zeatin biosynthesis, flavonoid biosynthesis, and plant hormone signal transduction were significantly enriched in the list of the upregulated genes in ZJ compared to DS at stage III (Figure 4A). In terms of downregulated genes, we observed that the biosynthesis of carotenoid and the tyrosine metabolic pathway were significantly enriched (Figure 4B).

### 2.5. Differentially Abundant Metabolites (DAMs) in DS and ZJ Mulberry Genotypes

To adequately document the metabolic changes in ZJ (the anthocyanin-rich genotype) compared to DS at stage III, we performed a non-targeted metabolome analysis. The recorded results exhibited a clear separation in terms of the PCA analysis between the two genotypes (Figure 5A). Hence, the top first principal component depicted more than 75% variation in the metabolism between both mulberry genotypes (Figure 5A). Indeed, we found that 310 and 290 metabolites were down- and up-regulated, respectively, in DS versus ZJ at developmental stage III (Figure 5B; Appendix A). Interestingly, these metabolites strongly correlated with each other for both mulberry genotypes (Appendix A). The analysis performed on the top 20 KEGG terms of the DAMs, detected based on a comparative study between the two mulberry genotypes at stage III, corroborated the presence of the following pathways: taste transduction, citrate cycle, arginine biosynthesis, pyruvate metabolism, and glyoxylate and dicarboxylate metabolism (Figure 5C). Additionally, the top 10% DAMs were found to be related to lignin metabolism, including arginine biosynthesis, glyoxylate and dicarboxylate metabolism, and ABC transporter (Figure 5C). In this regard, we identified several metabolites associated with anthocyanin metabolism, including naringenin, naringenin-7-O-glucoside, phloretin, procyanidin A1, kaempferol, kaempferol 3-O-rutinoside, quercetin, 3,4-dihydroxymandelic acid, cyanidin 3-glucoside, L-phenylalanine, and trans-cinnamate. These identified metabolites were found to be more abundant in the ZJ genotype than in the DS one (Figure 5D; Appendix A).

### 2.6. Ethylene-Induced ERF Gene Family Expression and Anthocyanin Accumulation

The abundances of the genes involved in the anthocyanin biosynthesis pathway, such as flavanone 3-hydroxylase, flavonol synthase, and arogenate dehydrogenase 1, were significantly enhanced in the presence of ethylene (Appendix A). In terms of the regulatory genes responsible for anthocyanin biosynthesis (as mentioned above by the upregulated genes based on the KEGG analysis), the plant hormone signal transduction was also significantly enriched (Figure 4A). Ethylene accumulated significantly, especially in ZJ at the later stages of the fruit ripening process (Figure 6A). Notably, 104 annotated genes related to the ethylene responses (Appendix A) showed different relationships with the anthocyanin dynamic content across the different post-flowering stages in the two mulberry genotypes, where the PCC ranged between −0.8 and 0.93 (Figure 6B).

Most of the ERF genes displayed a negative correlation with anthocyanin content changes (Figure 6C). Thus, the ethylene treatments exhibited a substantial improvement in the mulberry fruit ripening process and anthocyanin accumulation, especially at the concentration of 300 mg L^−1^ (Figure 6D,E). The expression profiling of four extremely upregulated *ERF* genes, i.e., LOC21390626, LOC21396325, LOC21397596, and LOC21406765, was further validated by q-RT-PCR (Figure 6F,I). The results revealed that the expression levels of the four *ERF* genes gradually increased following ethylene treatment at 300 mg L^−1^ (Figure 6F–I).

### 2.7. Anthocyanin Over-Accumulates in ZJ Genotype

To concretely understand the molecular mechanism behind the anthocyanin over-accumulation in ZJ compared to DS, we evaluated the expression levels of anthocyanin biosynthesis-related genes (e.g., flavanone-4-reductase, tyrosine kinase, cinnamoyl-CoA reductase 1, and flavanone 3-hydroxylase, each of which is known to be involved in the anthocyanin biosynthetic pathway) in both mulberry genotypes at stage III using q-RT-PCR (Figure 6A; Appendix A). Our results show that most of the flavonol biosynthetic-related genes were upregulated by 2 to 6 times in the ZJ genotype, such as flavanone 4-reductase (LOC21383800) and tyrosine kinase (LOC21384694), whereas genes related to salicylic and tyrosine metabolic pathways were slightly downregulated by just 1-fold in ZJ compared to DS (Figure 7A; Appendix A). Strikingly, the strong correlation between the expression of the genes involved in the anthocyanin biosynthetic pathway and the anthocyanin content for each mulberry genotype revealed the crucial role of these genes, identified based on the metabolite profiling assessed at stage III, in controlling anthocyanin accumulation in mulberry (Figure 7B,C; Appendix A). Notably, the regulatory genes involved in MBW complexes, such as the expression of *MYBA*, *F3H,* and *bHLH3,* were found to be significantly enhanced in ZJ compared to DS (Figure 7B; Appendix A). In addition, several other upregulated genes were identified, including the phenylalanine ammonia lyase class (*PHE1*), receptor protein tyrosine kinase (*CEPR2*), flavanone 4 reductase (*F4R*), flavanone 3 hydroxylase (*FS2*), and cinnamoyl CoA reductase 1 (*F4R1*) (Figure 7B,C; Appendix A). In contrast, the most significantly downregulated genes were salicylate carboxy-methyl-transferase (*SALIC*), tyrosine protein kinase (*Abl*), protein-tyrosine phosphatase (*IBR5*), salicylic acid binding protein 2 (*SALIC2*), protein tyrosine phosphatase MKP1 (*MKP1*), and tubulin tyrosine ligase like protein 12 (*TYRK12*) (Figure 7B,C; Appendix A).

In the same context, some *ERF* genes were found to be strongly expressed following ethylene treatment, including *ERF3* (LOC21397596) and *ERF5* (LOC21406765). Specifically, the expression level of the latter gene (*ERF5*) was enhanced by more than 10 times in ZJ compared to that in DS at stage III (Appendix A). This suggests that this gene is somehow involved in the higher anthocyanin content in ZJ. Overall, it seems that the upregulated *ERF* genes promote the metabolite accumulation, and thus, induce the anthocyanin metabolic pathway in both mulberry genotypes, but to a higher extend in ZJ than in DS (Figure 7D). Indeed, all the implicated genes in the main anthocyanin biosynthetic pathway from chorismate to anthocyanin were upregulated by up to 6 fold in ZJ vs. DS (Figure 7D). Two other pathways exist, i.e., the transition of chorismate to tyrosine and that of cinnamic acid to salicylic acid. Consequently, the genes involved in these pathways were found to be dramatically inhibited (Figure 7D). These recurrent and differential changes in the metabolic pathways between the two mulberry genotypes could explain, to a certain extent, the greater anthocyanin accumulation in the ZJ fruits compared to DS, as assessed at stage III (Figure 1C). This might also corroborate the differences in the fruit pigments between ZJ (darker color) and DS.

To confirm whether ERF5 binds to the motif in the promoter regions of *MYBA* in MBW complex and *F3H*, we performed both EMSA and Luciferase assays. The results suggested that ERF5 binds well to the ‘GCCGAC‘ motif located on the promoter regions of both *MYBA* and *F3H* (Figure 8A,B). ERF5 tends to enhance the expression of *MYBA* and *F3H* through in vivo transcription activation in tobacco leaves (Figure 8C,D). Collectively, the transcriptional activation exerted by ERF5 on both *F3H* and *MYBA* revealed the positive regulatory effects of ethylene on anthocyanin accumulation in mulberry fruits.

## 3. Discussion

The crude extracts of the purple-colored mulberry fruits at the ripening stage contained large amounts of anthocyanin. Many AP2/ERFs complexes are involved in the regulation of plant hormones, abscisic acid (ABA), and ethylene biosynthesis pathways to activate ABA and ethylene responsive genes [15], thereby promoting fruit ripening and anthocyanin accumulation. However, a relatively limited number of studies have focused on the regulatory mechanism of the *ERF* gene family in the anthocyanin biosynthetic pathway [16,17]. In this study, we performed an integrative analysis of the metabolome and transcriptome in two mulberry genotypes, characterized by their varying anthocyanin contents, especially in the final fruit developmental stages (stages III and IV). Our major findings revealed that ERF5 binds to the ‘GCCGAC’ motif located at the promoter regions of *MYBA* and *F3H* and transcriptionally activates the anthocyanin accumulation. Notably, we found a novel gene (*ERF5*) that positively regulates the expression of some genes involved in anthocyanin metabolism in mulberry. We debated the involvement of the casual *ERF* genes and the associated biosynthetic pathways in the regulatory mechanism of the anthocyanin accumulation in mulberry fruits.

### 3.1. Flavonoids Biosynthesis Enrichment in Anthocyanin-Abundant Genotypes

Anthocyanins belong to a large group of flavonols [18]. Thus, a huge natural variation exists in the anthocyanin levels of fruits of different mulberry genotypes [19]. Notably, the two mulberry genotypes used in this study showed a significant difference (*p* < 0.01) in the anthocyanin content at different post-flowering stages (Figure 1C). Thus, our KEGG analysis results, performed on the transcriptomes data, revealed that flavonoid biosynthesis-related secondary metabolites were highly accumulated in the fruits of the ZJ mulberry genotype (Figure 4A). Consistently, several genes involved mainly in the anthocyanin biosynthetic pathway were found to be substantially upregulated in ZJ compared to DS (Appendix A) [11,20], for instance, flavone-3-hydroxylase (LOC21396296 and LOC21403067), cinnamoyl-CoA reductase 1 (LOC21396683 and LOC21407528), and chalcone isomerase (LOC21394508 and LOC112095175) (Appendix A). In this context, our results are in agreement with the findings reported earlier [21]. Collectively, this confirms the higher anthocyanin level of ZJ compared to DS.

### 3.2. The Metabolome Profiling Concords well with the Anthocyanin Dynamic Changes in Mulberry

To properly comprehend the relationship between the metabolite profiling and anthocyanin dynamic changes in mulberry fruits, an extensive and reliable metabolic analysis related to anthocyanin metabolism was performed. In this regard, several substrates related to flavonol glycosides metabolism, such as naringenin, naringenin-7-O-Glucoside, phloretin, procyanidin A1, kaempferol 3-O-glucoside (astragalin), and quercetin 3-O-glucoside (isoquercitrin), were found to be substantially accumulated in ZJ compared to DS (Figure 5D; Appendix A). In addition, a considerable accumulation of phenylalanine, an intermediate substrate between the primary and secondary metabolisms, was observed in ZJ fruits [22]. Thus, some phenylalanine biosynthesis-related genes were highly upregulated in ZJ compared to DS (Appendix A). This is in line with the findings reported previously by Huang and coworkers regarding the anthocyanin abundant genotype, *Morus atropurpurea* Roxb [23].

Thus, in investigating an alternative pathway contributing to phenylalanine biosynthesis, we found that most of the genes involved in the tyrosine biosynthetic pathway were downregulated by up to 2 times in ZJ compared to DS (Figure 7A,D). This is very likely ascribable to the competitive relationship between the phenylalanine and tyrosine biosynthetic pathways, where both can be synthesized from chorismate, the final product of the shikimate pathway [24]. Interestingly, it seems that the phenylalanine accumulation in ZJ constitutes an effective precursor for cinnamic acids and/or chalcones, which might be the reason behind the anthocyanin biosynthesis excess (anthocyanin over-accumulation) in the ZJ mulberry genotype (Figure 6C). The consumption of this mulberry fruit in a basic diet would certainly help to maintain a high level of anti-inflammatory activity for humans, reinforcing the antioxidant activity of the organism [25,26].

### 3.3. Ethylene Promotes Anthocyanin Accumulation through ERF Gene Upregulation

The anthocyanin synthesis is mainly controlled genetically by certain regulatory genes, as well as by some plant hormones, such as ethylene [11]. Herein, most of the reported anthocyanin biosynthesis regulatory genes are included in a network of three transcription factors (TFs): MYB (v-myb avian myeloblastosis viral oncogene homolog) protein, bHLH (basic helix-loop-helix) protein, and WD40 (WD-40 has a scaffolding function) protein [11]. In this regard, we showed that some *ERF* genes, such as *ERF3* (LOC21397596) and *ERF5* (LOC21406765), are embedded in the same gene regulatory network, with MYBA and bHLH3 monitoring the anthocyanin biosynthetic pathway (Figure 7B; Appendix A). MYBs play a dual function (acting simultaneously as activators and repressors) in the anthocyanin biosynthetic process [27]. In this respect, it has been reported that the combination of *MYBA* and *bHLH3* activates the expression of anthocyanin biosynthetic genes including *F3H* and *5GT* [11]. This suggests that the *ERF5* gene could also participate in the anthocyanin biosynthesis regulation process. In this context, we found that the expression pattern of some *ERF* genes, especially ERF5, was substantially enhanced, with a PCC of 0.88, relative to the anthocyanin accumulation (Figure 7B; Appendix A). This is in line with what the authors observed in a recent study conducted on *Arabidopsis thaliana* [28]. In agreement with our current findings, an earlier observation revealed that the *ERF* genes can promote anthocyanin accumulation in apple fruit under stressful conditions [17].

Generally, the expression level of *ERF5* appears to be stimulated by ethylene treatment. In line with this, expression was found to be strongly enhanced in the anthocyanin rich mulberry genotype, ZJ (Appendix A). The plant hormonal (ethylene) stimulation of the *ERF5* gene facilitates the conversion of the shikimate, originating from the CBB cycle, into chorismate, which, in turn, can be conjugated to prephenate, phenylanine, cinnamic acids, chalcones, naringenin, dihydrokaempferol, and cyaniding, before producing anthocyanin in the final step (Figure 7D). Hence, it seems that the enhancement in anthocyanin production, and thereby its accumulation, may be attributed to the transcriptional activation of ERF5 upon the expression of *MYBA* from the MBW regulatory network as well as *F3H* (Figure 8). In two other reactions, i.e., the conversion of chorismate to tyrosine and the conversion of cinnamic acid to salicylic acid, the involved genes were found to be dramatically inhibited (Figure 7D). These dynamic changes explain the greater accumulation of anthocyanin in ZJ fruits if compared to DS based on stage-III estimations.

## 4. Material and Methods

### 4.1. Plant Material and Growth Condition

Two mulberry fruit samples, Zijing (ZJ) and Dashi (DS), were grown in the experimental field of the Hubei Academy of Agricultural Sciences (HAAS) (30.44° N, 114.3° E) in February 2016. The fruits were collected in April 2020. The average local temperature was 21.36 °C during the growth season. Tender mulberry fruits were collected from the top of the branches on 5, 18, 27 and 31 days after flowering (DAF). The four DAF were correspondingly named post-flowering stages I, II, III and IV, respectively.

The collected samples were used for various anthocyanin measurements. The fruit samples of the two mulberry genotypes used for q-RT-qPCR analysis and non-targeted metabolism were collected from a different tree than that used for the anthocyanin measurements. The samples were collected and immediately frozen in liquid nitrogen, and then stored at −80 °C until subsequent use. Notably, the fresh mulberry fruits were collected at different post-flowering stages from the same position of each branch. Each sample contained ten mulberry fruits. The samples were also collected to determine the anthocyanin content, fruit firmness and the soluble solids content (SSC).

### 4.2. Physiological Measurements

#### 4.2.1. Determination of Anthocyanin Content

With slight modifications, the anthocyanin content was measured at a wavelength of 657 nm, as previously reported [29]. The anthocyanin was extracted at 4 °C in the dark for 24 h from 0.05 g mulberry fruits in 2 mL of methanol:HCl (99:1, *v*/*v*) solution [30]. Using a UV-Vis 2450 spectrophotometer, the absorption spectra at 420–700 nm of the anthocyanin crude extracts were compared to the blank of methanol:HCl (99:1, *v*/*v*) (Shimadzu, Tokyo, Japan). The pigment content was calculated as cyanidin 3-glucoside, using an extinction coefficient of 29,600 L cm^−1^ mg^−1^ and a molecular weight of 448.8.

#### 4.2.2. Measurements of Fruit Solidity and Soluble Solid Content (SSC)

The fruit solidity and SSC were calculated as described earlier [31]. The fruit firmness was assessed in both mulberry genotypes (DS and ZJ) by a CT_3_ texture analyzer (3375 North Delaware Street, Chandler, AZ, USA) with a 2-mm diameter probe. The probe was pressed against ten independent fruits to a depth of 3 mm. In addition, 200 mL of pressed samples from the mulberry fruits were collected to determine the SSC amount using a PAL-1 refractometer (Minato-ku, Tokyo 105-0011, Japan).

#### 4.2.3. Ethylene Determination

The ethylene content in different mulberry genotypes was determined with a gas chromatograph (GC)-based method, as described previously [32]. Briefly, a 0.5-mL sample of headspace gas was injected into a portable GC (Photovac 10A10; Perkin Elmer, Waltham, MA) at ambient temperature (23 °C). The GC was equipped with a photo-ionization detector (PID) and a 30.0 × 0.125-inch 80/100-mesh activated alumina column. The carrier gas was nitrogen at 15mL min^−1^. The ethylene production rates were expressed in nanograms per fresh weight.

### 4.3. Transcriptome Analysis

#### 4.3.1. RNA Extraction

The total RNA was extracted from the fruit samples of the two mulberry genotypes using TRIzol Reagent, according to the manufacturer’s instructions (Invitrogen, San Diego, CA, USA), and the genomic DNA was removed with DNase I (TaKara, Forster City, CA, USA) as previously documented [33]. The RNA quality was then checked by a 2100 Bioanalyser (Agilent, San Diego, CA, USA) and quantified using the ND-2000 (NanoDrop Technologies, Irvine, CA, USA). To construct the sequencing library, only high-quality RNA samples (OD_260_/OD_280_ = 1.8~2.2, OD_260_/OD_230_ ≥ 2.0, RIN ≥ 6.5, 28S:18S ≥ 1.0, >10 μg) were considered for RNA-Seq processing.

#### 4.3.2. RNA-Sequencing (RNA-Seq)

The total RNA extracts from several samples of mulberry fruits were prepared independently. Equivalent volumes of RNA taken from all samples were considered to construct the RNA-Seq libraries. The VAHTSTM mRNA-Seq v.2 Library Prep package for Illumina (Vazyme Biotech Co., Nanjing, China) was used to build the library. Using poly-T oligo-conjugated magnetic beads, approximately 1 mg of total RNA was purified, and the cleaved RNA was used for the cDNA synthesis with random primers. The purified cDNA fragments were blunted at the ends and a poly-A tail was applied before the adapter ligation. PCR amplification of cDNA fragments was carried out using PCR specific primers and a specific amplification Mix (Vazyme Biotech Co., Ltd., Nanjing, China). Ultimately, the reaction products were purified using a Qubit 2.0 DNA kit and the double-stranded cDNA was used for HiSeq 2500 paired-ends sequencing.

#### 4.3.3. De Novo Transcriptome Data Processing and Assembly

A Perl script was written to remove the vector sequences and PolyA (T) tails from the raw sequence data using the cut-adapt software, version 1.2.1 (http://www.pypi.python.org/pypi/cutadapt/1.2.1, accessed on 18 May 2020). The low-quality reads (35-bp) were removed using the Prin-seq software, version 0.19.5 (http://www.prinseq.sourceforge.net/, accessed on 18 May 2020). Then, the reads were merged after discarding repeats. The high-quality reads were assembled using the Trinity software, version r20140717 (http://www.trinityrnaseq.github.io/, accessed on 18 May 2020), to construct a unique transcript [34].

#### 4.3.4. Identification of Differentially Expressed Genes (DEGs)

The genes with differential expression levels between the two cDNA libraries of the two mulberry genotypes were identified using the DESeq software (http://www.bioconductor.org/, accessed on 20 June 2020) according to a previously reported method [35]. The DEGs were defined as the genes showing differential expressions at a *p*-value < 0.01 and an expression ratio (ZJ/DS) higher than 2 (FC > 2). To identify the DEGs between the two different samples, the expression level of each transcript was calculated according to fragments per kilo base of exon model per million reads mapped (FPKM) method, which considers the gene length for normalization to be suitable for the sequencing protocols. In this context, it has been reported that the reads sequencing process depends on the gene length [36].

#### 4.3.5. Functional Annotation and Genes Classification

The identified genes were compared using the basic local alignment search tool (BLAST) of the National Center for Biotechnology Information non-redundant nucleotides (NT, ftp://ftp.ncbi.nlm.nih.gov/blast/db/nt.tar.gz, accessed on 20 June 2020, via BlastN) and non-redundant proteins (NR, ftp://ftp.ncbi.nlm.nih.gov/blast/db/nr.tar.gz, accessed on 20 June 2020, by BlastX databases) websites. A SWISS-PROT using BlastX was used for gene annotations and classification. The clusters of ortholog groups of proteins database (COG; E-values 1 × 10^−10^, using rpsBlast), (KEGG, release 58; E-values 1 × 10^−10^) and an InterProScan Release 36.0 annotated protein domains and functional assignments were mapped onto Gene Ontology (GO, http://www.geneontology.org/, accessed on 20 June 2020) using the BlastX algorithm.

### 4.4. Quantitative Real-Time (q-RT-PCR) Analysis of Differentially Expressed Unigenes

In total, 24 DEGs were used for the q-RT-PCR analysis. The primer sequences used for the q-RT-PCR amplification reaction are listed in Appendix A. The total RNA was extracted from the fruits of the two mulberry genotypes (DS and ZJ) at different stages (I, II, III and IV) and was reverse-transcribed using the RevertAid first Strand cDNA Synthesis Kit (Thermo Fisher Scientific, Waltham, MA, USA). The q-RT-PCR experiments were conducted using a fast start essential DNA Green Master kit, and the Cq values were determined with a Light Cycle 96 Real-Time PCR system (Roche, Basel, Switzerland). The primers were designed using the Primer Premier 5.0 software (Premier Biosoft Ltd., Palo Alto, CA, USA) and were obtained from Sangon Biotech Company (Shanghai, China). The q-RT-PCR program consists of a reverse transcription step at 48 °C for 30 min and a Taq polymerase activation step at 95 °C for 30 s, followed by PCR: 45 cycles at 95 °C for 15 s, 61 °C for 20 s, and 72 °C for 30 s, and finally, a melting cycle. The 2^−ΔΔCT^ method [37] was used to calculate the relative changes in the target gene expression as assessed by q-RT-PCR analysis, relative to the mulberry *Actin* gene as a reference. All q-RT-PCR experiments were carried out based on three biological replicates.

### 4.5. LC-MS/MS Metabolism Analysis and Metabolites Identification

A non-targeted metabolic profile in the fruits of both mulberry genotypes, collected at four post-flowering stages, was prepared based on the LC-MS/MS (Q Exactive, Thermo Scientific) procedure, as earlier described [11]. Briefly, ~2.5 mg fruit samples collected from DS and ZJ mulberry genotypes at the different stages mentioned above were sampled in 2 mL Eppendorf tubes containing pre-cooled metal beads and were then immediately stored in liquid nitrogen. The samples were first extracted with a ball mill at 30 Hz for 5 min; then, the extracted powder was dissolved in a 1.5-mL methanol/chloroform mixture and incubated at −20 °C for 5 h. Thereafter, the mixture was centrifuged at 2000× *g* and 4 °C for 10 min and then filtered with 0.43 μm organic phase medium (GE Healthcare, 6789-0404).

The metabolomic analysis was performed using the metabolon software (Durham, NC, USA). The sample components were identified by comparing the retention time and mass spectra with those of the reference metabolites (one by one). Regarding the identification of metabolic compounds in each sample, the mass spectra with the entries of the mass spectra libraries NIST02 and the Golm metabolome database were considered (http://csbdb.mpimp-golm.mpg.de/csbdb/gmd/gmd.html, accessed on 20 June 2020).

### 4.6. Electrophoresis Mobility Shift Assay (EMSA) Experiments

The EMSA procedure was carried out as described in detail by a member of our team [38]. Briefly, the cDNA of the *ERF5* (LOC21406765) gene (Appendix A) was cloned into the expression vector pET51b (+) between the restriction enzyme sites *Bam*HI and *Sac*I via the homologous recombination method (C112, Vazyme Biotech Co., Ltd., Building C2 Hongfeng Tech Park Kechuang Road, Nanjing, China). The cDNA of *ERF5* was linked with a Strep II tag at the 5′- end and His × 10 at the 3′- end (StrepII::*ERF5*::10× His) in the pet51b (+) plasmid. Protein was expressed in competent *E. coli* cells (strain BL21) and induced using isopropyl-β-d-thiogalactopyranoside (IPTG; 50 mM). To design ERF binding DNA probes, the ‘GCCGAC’ motif was sought within 3000 bp upstream of the ATG in the promoter regions of the genes *MYBA* (LOC112093091) and *F3H* (LOC21403067). Therefore, to synthesize the fluorescent probes, the DNA amplifying primers were linked with CY5 fluorescent labeling at the 5′ end. The amplified motifs (60–100 bp) were then purified as probes. The purified ERF5 protein (200 ng) was incubated with the fluorescent probes for 30 min at 25 °C in an EMSA buffer [100 mM Tris (pH 7.9), 25% glycerol, 0.2 μg μL^−1^ BSA] and a binding buffer (5× EMSA buffer, 1 M MgCl_2_, 0.5 M DTT, and 1 mg m^−1^ salmon sperm DNA). For the competition test, the mutated probes, synthesized by Sango Private Limited (Shanghai, China), were added into the binding reaction. The primers used to amplify the motif sequence are listed in Appendix A. Each reaction was loaded onto a 6.5% native polyacrylamide gel with 0.5× TBE buffer and was run at 100 V for 1 h. The gel was then exposed to X-ray film to produce images. The experiment was performed twice and the results were found to be reproducible.

### 4.7. Luciferase Assays

For the analysis of the *MYBA* and *F3H* promoter in response to anthocyanin accumulation through the transcriptional activation of ERF5, exactly 2.0 kbp DNA fragments upstream of the two genes coding region (CDs) were amplified with a base pair “AC” insertion into pGreenII 0800-LUC vector to generate pro*MYBA*:LUC and proF3H:LUC, respectively. The primers used for the PCR amplification are given in Appendix A. Ten-day-old Arabidopsis seedlings, grown in a growth chamber with a controlled temperature of 23 °C/22 °C (day/night), were prepared for protoplast transient localization, as described earlier [39]. Accordingly, all vectors were transformed into Arabidopsis protoplasts. The activities of firefly luciferase (LUC) and *Renilla*
*luciferase* (REN) were examined using a Dual-Luciferase Reporter Assay System kit (Promega, E1960). The LUC/REN ratio was calculated and considered as the luciferase relative activity. For each vector, at least three replicates were used to evaluate the ERF5 binding affinity.

### 4.8. Statistical Analysis

All data presented in this manuscript are expressed as the mean values and the standard deviation (±SD). Data were evaluated for statistically significant difference using a Student’s *t*-test. The SPSS software version 13.0 (IBM SPSS, Armonk, NY, USA) was employed for the statistical analyses and differences were considered to be statistically significant at a *p*-value < 0.05. The correlation analysis was performed using the GraphPad Prism software, version 5.0 (San Diego, CA, USA).

## 5. Conclusions

In summary, we observed substantial differences in anthocyanin contents between the two studied mulberry genotypes (ZJ and DS). The dynamics of global metabolites and genes were reported in both mulberry fruits at different post-flowering stages. Specifically, 1035 upregulated genes in ZJ compared to DS were identified, including anthocyanin biosynthesis-related genes, known as MBW complex genes and some *ERF* ones. The promoted effects of ethylene on anthocyanin accumulation are ascribable to the transcriptional activation of ERF5 on the *F3H* and *MYBA* genes. This work improves our knowledge about the regulation of the anthocyanin biosynthesis process, as mediated by ethylene in mature mulberry fruits.

## Figures and Tables

**Figure 1 ijms-23-07615-f001:**
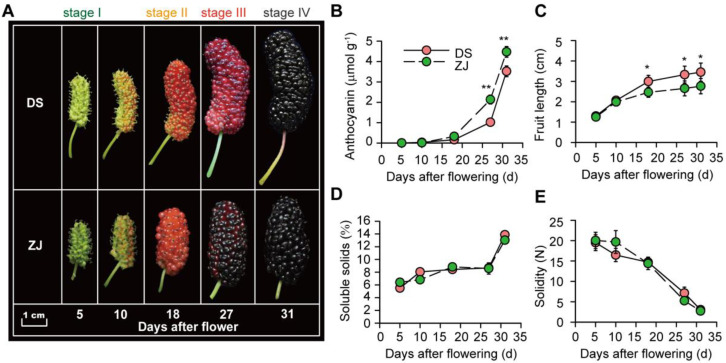
Mulberry fruit morphology and physiological traits of Dashi (DS) and Zijin (ZJ) genotypes at different post-flowering stages. (**A**) Morphological appearance of the fruits of two mulberry genotypes. (**B**–**E**) Anthocyanin content, fruit length, soluble solids, and solidity for two mulberry types at various post-flowering stages. In panel A, the white horizontal bar represents the scaling bar of 1 cm length. For panels (**B**–**E**), each data point of the curves represents the mean of four independent replicates (±SE). Symbols “*” and “**”, represent significant differences between DS and ZJ at a *p*-value < 0.05 and 0.01, respectively, based on a Student’s *t*-test.

**Figure 2 ijms-23-07615-f002:**
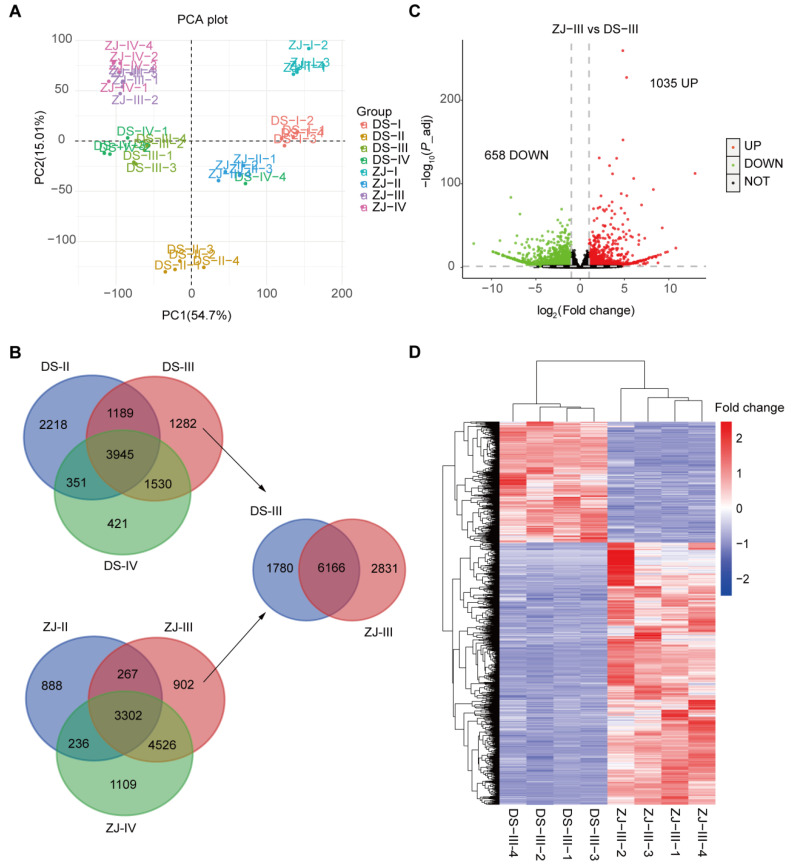
Principal component analysis (PCA) and differentially expressed genes (DEGs) performed on various mulberry fruits samples obtained from four different post-flowering stages (**A**) PCA performed for DS and ZJ at four different post-flowering stages. (**B**) Venn diagram representing the overlapped genes for the two mulberry genotypes (DS and ZJ) at different post-flowering stages relative to the first one (stage I) and the third one (stage III). DS2-1, DS3-1, DS4-1 and ZJ2-1, ZJ3-1, and ZJ4-1 represent the different stages for both mulberry genotypes. (**C**) Volcano plot displaying the DEGs for the two mulberry genotypes across the different post-flowering stages. (**D**) Heatmap of the top 20% DEGs for the two mulberry genotypes atfruit developmental stage III.

**Figure 3 ijms-23-07615-f003:**
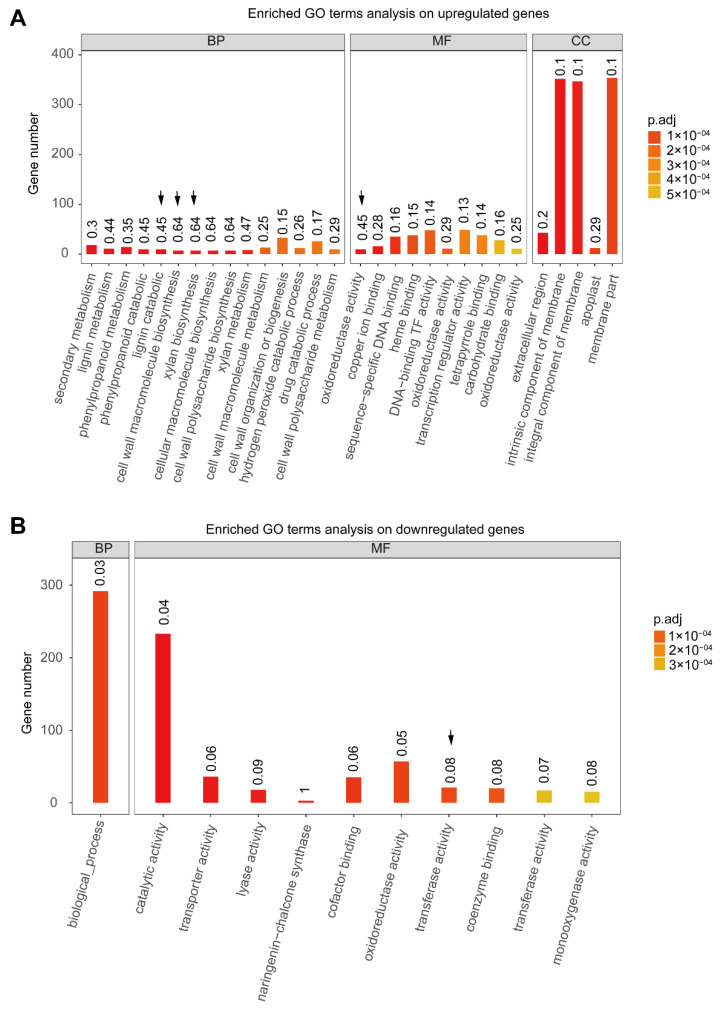
Top-20 enriched gene ontology (GO) in terms of up- and down-regulation of the expressed genes associated with anthocyanin abundance in mulberry genotype ZJ (high anthocyanin level) compared to DS (low anthocyanin level) at stage III. (**A**,**B**) represent the up- and down-regulated genes related to anthocyanin biosynthesis enrichment in mulberry genotype ZJ relative to DS at stage III. The downward arrows represent the key highlighted biological pathways in this study.

**Figure 4 ijms-23-07615-f004:**
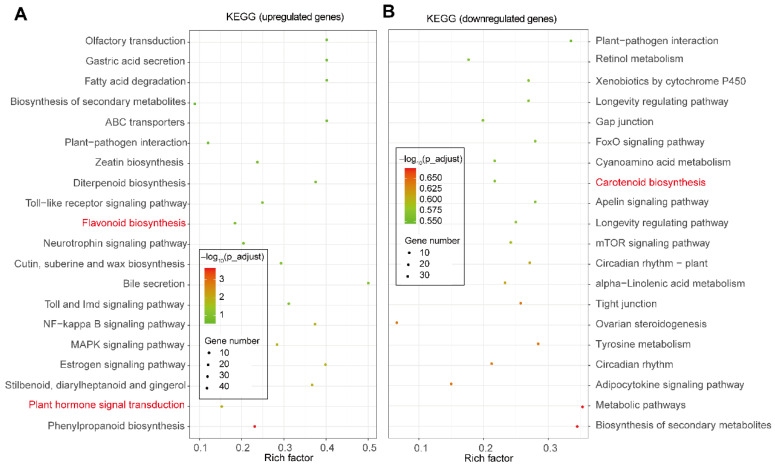
Top-20 enriched KEGG in terms of up- and down-regulation of the expressed genes monitoring the anthocyanin abundance in the fruit of mulberry genotype ZJ relative to DS at stage III. (**A**,**B**) represent the up- (**A**) and down-regulated (**B**) genes associated with anthocyanin biosynthesis enrichment in mulberry genotype ZJ relative to DS at stage III. The red fonts display the key highlighted metabolic pathways in this study.

**Figure 5 ijms-23-07615-f005:**
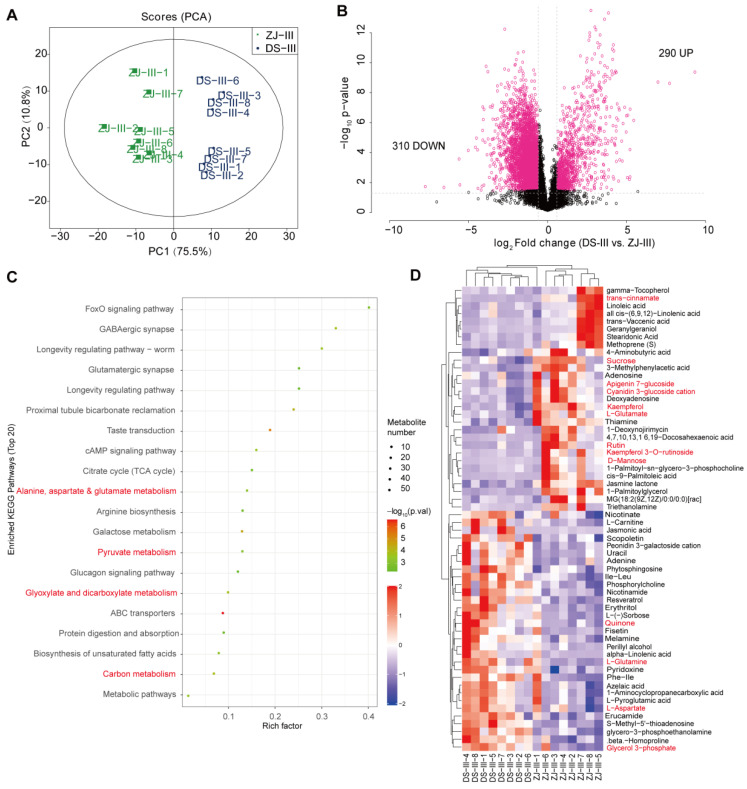
The differentially abundant metabolites (DAMs) analysis performed using a non-targeted metabolome analysis in two mulberry genotypes (ZJ and DS) at stage III (27 days) after flowering. (**A**) PCA on metabolism in the ZJ and DS genotypes. (**B**) Volcano plot depicting the DAMs in ZJ and DS. Pink and black dots represent significant and not significant metabolites. (**C**) Top-20 KEGG terms on DAM between the two mulberry genotypes. (**D**) Comparison of DAMs between the two mulberry genotypes at stage III. Red words in panels (**C**,**D**) stand for metabolites involved in carbon assimilation metabolic pathways and flavonoid biosynthetic pathways, respectively.

**Figure 6 ijms-23-07615-f006:**
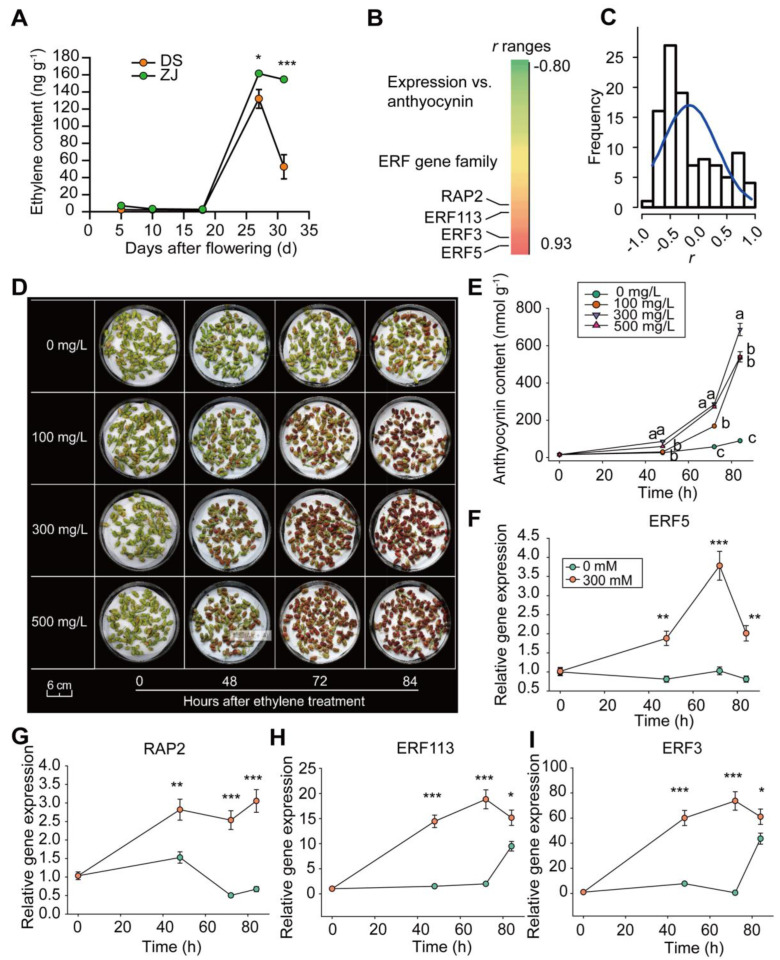
Ethylene promotes anthocyanin accumulation by upregulating ERF genes in mulberry. (**A**) Ethylene content in both mulberry genotypes at different post-flowering stages. (**B**) Pearson correlation coefficient, PCC (*r*) between the anthocyanin content and the expressions of the ERF gene family in DS and ZJ at different post-flowering stages. (**C**) Distribution of *r* defined as the correlation between anthocyanin content and ERF gene expression. (**D**) Images of ethylene-treated fruits of ZJ in vitro for different durations. Ethylene concentrations ranged between 0 mg L^−1^ and 500 mg L^−1^. (**E**) Anthocyanin contents in ZJ fruits exposed to different ethylene concentrations for different durations. (**F**–**I**) Relative gene expression of different ERF genes in response to ethylene treatment of 300 mM for different durations. Data were normalized to the first data point for each time-point treatment. For panels (**A**,**E**–**I**), the data represent the mean values (*n* = 3) ± SE. For the same panels, significant differences among values were assessed based on the Student *t*-test. “*”, “**”, and “***” represent *p*-values < 0.05, 0.01 and 0.001, respectively. For panel (**E**), the different letters adjacent to the curves reflect significant differences level at a *p*-value < 0.05 based on one-way *ANOVA*.

**Figure 7 ijms-23-07615-f007:**
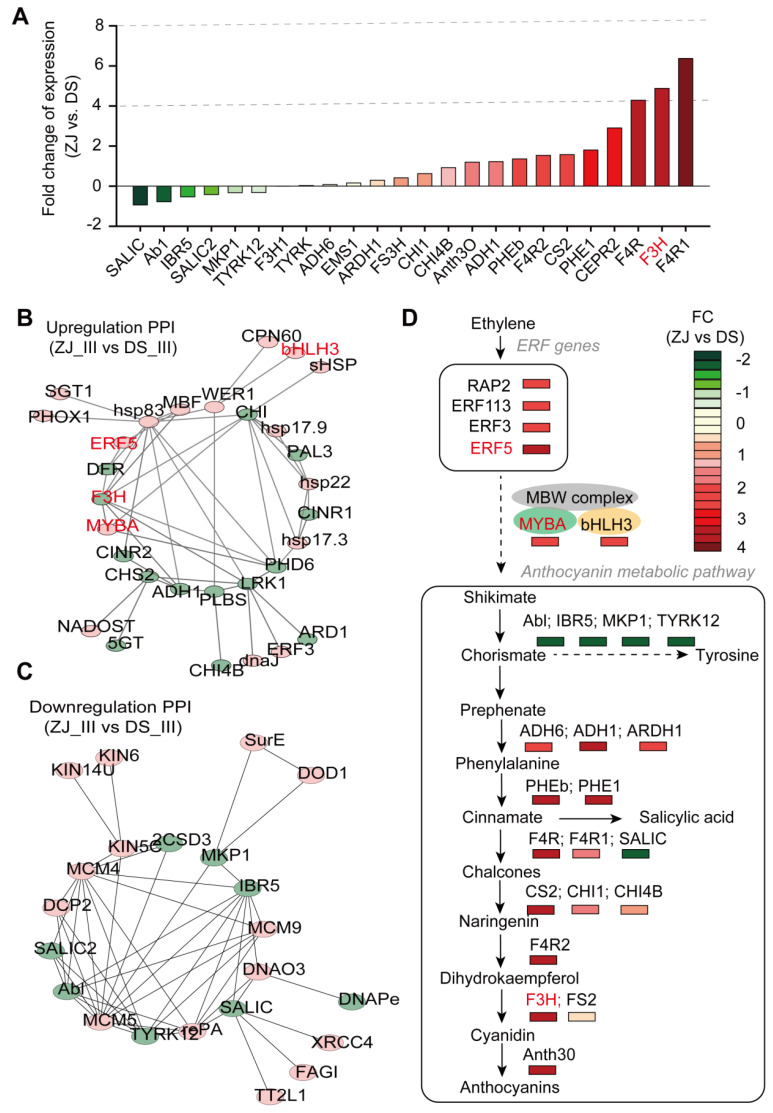
Potential regulatory network of ethylene-induced anthocyanin accumulation in both mulberry genotypes. (**A**) Percentage differences of gene expression in fruits of ZJ relative to DS at post-flowering stage III (27 days), expressed as fold changes (FC). Three technical replicates were performed on three independent biological samples for each gene represented in the histogram. The percentage differences expressed in FC of ZJ relative to DS are depicted in different colors. (**B**,**C**) Protein–protein interaction (PPI) network involving the differentially expressed genes (DEGs) related to the anthocyanin biosynthesis pathway and heat shock proteins (HSPs) gene family for both mulberry genotypes (ZJ and DS). The DEGs include the upregulated genes (**B**) and downregulated ones (**C**). The Detailed information about the genes is provided in Appendix A. The 22 genes related to anthocyanin biosynthesis are highlighted in green. (**D**) A simplified mechanistic model of the ethylene-induced anthocyanin accumulation in both mulberry genotypes to various extents thanks to the ERF. The scale bars with different colors represent the FC of gene expression in ZJ relative to DS at stage III (27 days).

**Figure 8 ijms-23-07615-f008:**
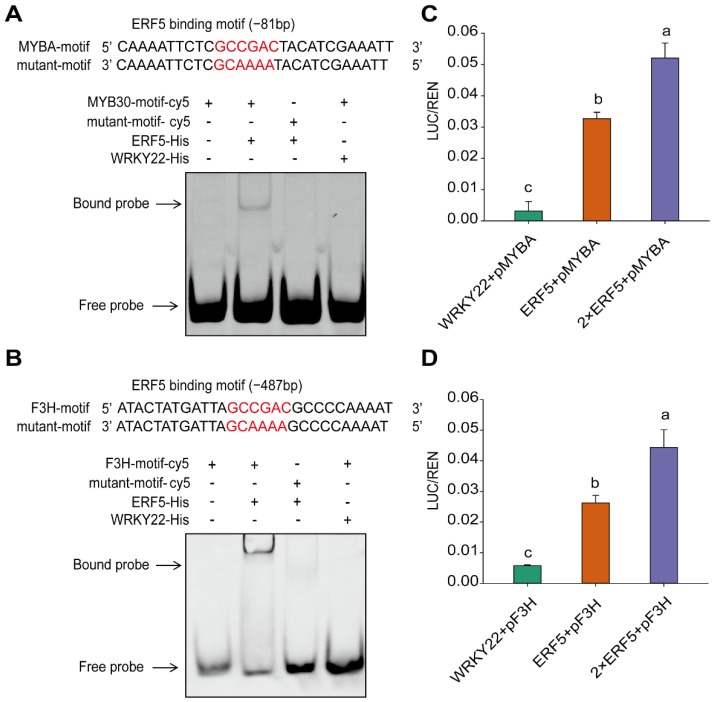
ERF5 promotes the gene expression of *MYBA* and *F3H* in mulberry fruits. (**A**,**B**) ERF5 binds to the promoter region of *MYBA* and *F3H,* as validated by electrophoresis mobility shift assay (EMSA). (**C**,**D**) ERF5 promotes gene expression of *MYBA* and *F3H* through a transcriptional activation process, as assessed based on luciferase assay. Different letters represent the significant level at *p* < 0.05 based on student *t*-test.

## Data Availability

All data are available in the manuscript or the Appendix A.

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
