# Peer review of "The Ethylene Response Factor ERF5 Regulates Anthocyanin Biosynthesis in ‘Zijin’ Mulberry Fruits by Interacting with MYBA and F3H Genes"

_ijms, 2022, doi:10.3390/ijms23147615_

Round 1

Reviewer 1 Report

This study by Mo et al. performed through transcriptomic and metabolic analysis to identify pathways and genes involved in fruit ripening in two mulberry genotypes. The data are solid and supported the conclusion that ERF5 regulates anthocyanin biosynthesis by interacting with MYBA and F3H genes. However, I find this paper was poorly written and presented many grammar mistakes, which affects the soundness and readability of the manuscript as a scientific article. I recommend the authors to consult with professional English writing agency to proofread their writing carefully before publication.

Reviewer 2 Report

In the manuscript entitled «The ERF5 transcriptionally induces the MBY30 and F3H expressions and boosts the anthocyanin accumulation in mulberry fruits» the authors performed analyzes of transcriptomes and metabolism of fruits of two mulberry genotypes in order to study in more detail the biosynthesis of anthocyanins. This topic is interesting, and although many articles have been published on it, a large number of questions remain about the regulation of the biosynthesis of this class of secondary metabolites.

After reading this manuscript, I did not have any significant questions regarding the results. However, a big note about the English language. I recommend that the authors review the writing of the manuscript with the assistance of a specialist.

This work is worthy of publication in the IJMS journal, but after minor revision. I provide my comments below.

·         I wonder what is the current title of the work, which is indicated in the manuscript or indicated in the information on the site?

·         Indicate the coordinates of the plantation where the plants were grown. This will give a more detailed understanding of the growing conditions. It would also be good to indicate the date of collection of the fruit.

·         It would be good to describe in more detail the method for determining the content of anthocyanins in samples.

·         Lines 123-129. Indicate not only the name of the reagents and equipment manufacturers, but also their city and state.

·         Lines 123-129. of fruits of

·         Lines 36-40. This sentence is spelled wrong. Rewrite more accessible.

·         Lines 41.  such as, the nutritional. Extra comma after "as"

·         Lines 94.  Dashi (DS), WERE obtained. To correct

·         Lines 96.  Fruits WERE collected from. To correct

·         Line 110. «reported by….» Indicate the author of the work you are referring to.

·         Line 121. Was change were.

·         Line 154, 192. «reported by….» Indicate the author of the work you are referring to.

Reviewer 3 Report

Manuscript ID: ijms-1683546

In this paper, entitled “The ethylene response factor ERF5 regulates anthocyanin biosynthesis in ‘Zijin’ mulberry fruits by interacting with MYBA and F3H genes”, the Authors studied in detail the molecular mechanisms responsible for anthocyanin accumulation in mulberry (Morus alba L.) fruits by identification of genes involved in the regulation of anthocyanin biosynthesis. This research successfully identified ERF5 as a key factor interacting with MYBA and F3H for the regulation of anthocyanin biosynthesis in the fruit of the genotype ‘Zijin’ of mulberry (Morus alba L.).

Although this work is interesting, the results are not clearly presented so the paper has to be thoroughly modified. To improve the ms, English language has to be deeply revised, above all the correlation of verbal tenses and the consistent use of the present or past tense. Then, there are several typos and repetitions. All these deficiencies decrease the value of ms, despite its relevance. This reviewer will be very pleased to recommend it for publication after the suggested revisions.

Here are some specific comments:

ABSTRACT

L13-14: Anthocyanins are a group of similar but different molecules: please, use the plural “anthocyanins” instead of the singular “anthocyanin”, consequently change this sentence

L17-18: here and wherever it occurs, only use DAF or stage, not both.

L20,22,24: when possible use the simple past tense

L24: change “assays” to “assay”

INTRO

L34: change to “Mulberry (Morus alba L.) is a deciduous tree belonging to the Moraceae family.

L36: use not italics typing for “monophagous silkworm”

L36-42: the sentences do not have a clear meaning, please, rewrite more clearly

L36: check for “Mallotus leaves”

L43: see L13-14

L47-49: please, rewrite more clearly

L60-63: remove repetition of “recently… reported”

L68: check tense

L72: remove “obviously”

L78: add “analysis” after “metabolism”

L79: here, insert the names of mulberry genotypes ‘Zijin’ (ZJ) and ‘Dashi’ (DS)

L85: remove “working model displaying the”

M&M

L203: suggestion of removing “it is highly recommended to consider”, and adding in L205 “were considered” after “database”

RESULTS

L252: here and wherever it occurs, throughout the entire text, only use DAF or stage, not both.

Figure 1. Move the legend from Fig. 1C to Fig. 1B. Use “Days after flowering (daf)” for each axis “x” instead of “Time after flowering (d)”.

Figure 2. L306: remove “performed”

L327: check for English language, move “related” after “processes”

L332: check for English language, move “associated” after “pathways”

L354: here and wherever it occurs, throughout the entire text, only use DAF or stage, not both.

L354: change “metabolism” to ““metabolome analysis”

L365: fix the typo in “dicaboxylate”

Figure 2. L374: change “metabolism” to ““metabolome analysis”

L384-385: the description of methods for ethylene determination is lacking in M&M section, please fix

Figure 6A. see note of Figure 1. Use “Days after flowering (daf)” for axis “x” instead of “Time after flowering (d)”.

L400: fix the typo in “satges”

L403: fix the typo in “Anthyocynin”

L410: change the title of paragraph by using the answer to the title question.

L417: insert the main genes

L444: insert “in ZJ vs. DS” after “folds”

L444-445: check for English language

L444: as an alternative to what ?

L466: ERF5 does not bind MYBA and F3H proteins directly, but a specific motif of their promotor, please rewrite more clearly

L468: check for the correct motif “GCCGAC”

DISCUSSION

L480-481: check for English language, please rewrite more clearly

L485: change “metabolism” to ““metabolome”

L486: change “obvious” to “evident”

L491: change “biological” to “biosynthetic”

L494-495: see the comments to L13-14:

L509: change “metabolism” to ““metabolome”

L518: phenylalanine or tryptophan? or both ? or did you mean tyrosine instead of tryptophan ? please, specify more clearly

L533: please add this recent reference   https://doi.org/10.3390/molecules25010233

L565: remove full stop, and merge the two sentences by a comma

L568,572: see comments to L252
